# Structure of a cyanobacterial photosystem I surrounded by octadecameric IsiA antenna proteins

Fusamichi Akita[1,2,8✉], Ryo Nagao[1,8], Koji Kato[1,8], Yoshiki Nakajima[1], Makio Yokono[3], Yoshifumi Ueno[4], Takehiro Suzuki[5], Naoshi Dohmae[5], Jian-Ren Shen[1✉], Seiji Akimoto[4✉] & Naoyuki Miyazaki[6,7✉]

Iron-stress induced protein A (IsiA) is a chlorophyll-binding membrane-spanning protein in photosynthetic prokaryote cyanobacteria, and is associated with photosystem I (PSI) trimer cores, but its structural and functional significance in light harvesting remains unclear. Here we report a 2.7-Å resolution cryo-electron microscopic structure of a supercomplex between PSI core trimer and IsiA from a thermophilic cyanobacterium *Thermosynechococcus vulcanus*. The structure showed that 18 IsiA subunits form a closed ring surrounding a PSI trimer core. Detailed arrangement of pigments within the supercomplex, as well as molecular interactions between PSI and IsiA and among IsiAs, were resolved. Time-resolved fluorescence spectra of the PSI–IsiA supercomplex showed clear excitation-energy transfer from IsiA to PSI, strongly indicating that IsiA functions as an energy donor, but not an energy quencher, in the supercomplex. These structural and spectroscopic findings provide important insights into the excitation-energy-transfer and subunit assembly mechanisms in the PSI–IsiA supercomplex.

[1] Research Institute for Interdisciplinary Science and Graduate School of Natural Science and Technology, Okayama University, Okayama 700-8530, Japan. [2] Japan Science and Technology Agency, PRESTO, Saitama 332-0012, Japan. [3] Nippon Flour Mills Co., Ltd., Innovation Center, Kanagawa 243-0041, Japan. [4] Graduate School of Science, Kobe University, Hyogo 657-8501, Japan. [5] Biomolecular Characterization Unit, RIKEN Center for Sustainable Resource Science, Saitama 351-0198, Japan. [6] Life Science Center for Survival Dynamics, Tsukuba Advanced Research Alliance (TARA), University of Tsukuba, Ibaraki 305-8577, Japan. [7] Institute for Protein Research, Osaka University, Osaka 565-0871, Japan. [8]These authors contributed equally: Fusamichi Akita, Ryo Nagao, Koji Kato. ✉email: fusamichi_a@okayama-u.ac.jp; shen@okayama-u.ac.jp; akimoto@hawk.kobe-u.ac.jp; naomiyazaki@tara.tsukuba.ac.jp

Light-harvesting complexes (LHCs) are a family of pigment-proteins in photosynthetic organisms that play fundamental roles in harvesting solar energy and transferring them to the two photosystems, photosystem I (PSI) and II (PSII), where charge separation and electron transfer reactions are initiated[1]. Among the photosynthetic organisms, various LHCs have been developed to capture the solar energy under different light environments. In most plants and algae, LHCs are membrane-embedded protein complexes associated with the two photosystems, and the structures of various LHC-photosystem supercomplexes have been determined from a number of different species: PSI–LHCI from pea[2,3], PSI–LHCI from a red alga[4,5], PSI–LHCI from green algae[6–8], $C_2S_2$-type PSII–LHCII from spinach[9], $C_2S_2M_2$-type PSII–LHCII from Pea[10], and $C_2S_2M_2N_2$ ($C_2S_2M_2L_2$)-type PSII–LHCII from a green alga[11,12]. By contrast, cyanobacteria have evolved a completely different antenna system termed as phycobilisome, which is a huge water-soluble pigment–protein complex attached at the stromal surface of the photosystems[13]. However, some cyanobacteria possess membrane-embedded light-harvesting protein complexes, namely, prochlorophyte chlorophyll (Chl) $a/b$ protein (Pcb) and iron-stress-inducible A protein (IsiA)[14–20]. IsiA has six transmembrane helices with a structural similarity to the CP43 subunit in PSII[21,22], and is expressed under various stress conditions[23,24], especially under an iron starvation condition[19,25]. The IsiA protein is specifically associated with PSI, but not with PSII, through formations of ring structures around the PSI core. The IsiA ring can be either a single or double layer depending on the growth conditions and the species of cyanobacteria, although IsiA itself can also assemble into a single or double ring even in the absence of the PSI cores[26,27].

IsiA has been reported to play a role in donating energy to the PSI core in the PSI–IsiA supercomplex[28–31], whereas free IsiA is likely involved in energy quenching once IsiA is detached from PSI[31–34]. However, these spectroscopic results cannot exclude the possibility that energy quenching by IsiA may also occur in the PSI–IsiA supercomplex in a very early time region such as femtoseconds under physiological-temperature conditions. Recently, the overall architecture of PSI–IsiA has been determined at 3.5 Å resolution from a mesophilic cyanobacterium *Synechocystis* sp. PCC 6803 (hereafter designated as *S.* sp. PCC 6803, and PSI–IsiA from *S.* sp. PCC 6803 is designated as S_PSI–IsiA) by cryo-electron microscopy (cryo-EM)[35]. The structure showed that 18 IsiAs surround a PSI core trimer in a single ring organization, and implicated that IsiA may function in either energy harvesting or quenching[35].

To examine the structure and function of IsiA in a greater detail, we solved the structure of a PSI–IsiA supercomplex from a thermophilic cyanobacterium *Thermosynechococcus vulcanus* (*T. vulcanus*) by single-particle cryo-EM analysis at a much improved resolution of 2.7 Å, which revealed the excitation energy transfer (EET) pathway and detailed protein–protein and pigment–pigment interactions between PSI and IsiAs. We also performed time-resolved fluorescence (TRF) analysis of the PSI–IsiA supercomplex to examine its EET dynamics. Together with the spectroscopic observations, the PSI–IsiA structure from *T. vulcanus* (T_PSI–IsiA) provides important functional insights into how IsiA contributes to the EET events in the PSI–IsiA supercomplex.

## Results

**Structural determination and overall structure of PSI–IsiA.** We purified the PSI–IsiA supercomplex from *T. vulcanus* (Supplementary Fig. 1), cultured under an iron-starved condition as reported previously[27]. The purified PSI–IsiA retains all known cyanobacterial PSI subunits and the IsiA protein (Supplementary Fig. 1c). With this purified PSI–IsiA, we performed cryo-EM

single-particle analysis (Supplementary Fig. 2). Two-dimensional classification of the cryo-EM particles revealed that octadecameric IsiA proteins surround the trimeric PSI core entirely in a single ring (Supplementary Fig. 2b). The final cryo-EM map was reconstructed from 303,983 particles taken from 20,799 micrographs at 2.74 Å resolution (Supplementary Fig. 2c and Supplementary Table 1), which was estimated by the gold-standard FSC with a 0.143 cutoff (Supplementary Fig. 3a and Supplementary Table 1). The local resolution map was shown in Supplementary Fig. 3c. The peripheral regions of PSI–IsiA have relatively low resolution, resulting in rather high numbers of clashscore and poor rotamers in the final refined structure (Supplementary Table 1). However, the Chl and Bcr molecules, which were related to energy pathways, have well-defined densities, and therefore were completely assigned. The overall structure of the PSI–IsiA supercomplex has a size of 294 Å in diameter and 110 Å in thickness, and is composed of a trimeric PSI core surrounded by 18 IsiAs with a C3 symmetry (Fig. 1), namely one asymmetric unit contains a PSI monomer and six IsiAs. The six IsiAs in an asymmetric unit are named IsiA1–6 clock-wisely according to their positions relative to the PSI monomer (Fig. 1a, b).

Based on the 2.74 Å resolution cryo-EM map, an atomic model was built and refined (Fig. 1, Supplementary Fig. 4). In the final structure, the PSI–IsiA supercomplex contains 12 PSI core subunits (PsaA, PsaB, PsaC, PsaD, PsaE, PsaF, PsaI, PsaJ, PsaK, PsaL, PsaM, and PsaX), 18 IsiAs, 585 Chls, 138 β-carotenes (Bcr), 6 phylloquinones (Pqn), 9 phosphatidyl-glycerol (LHG), 3 distearoyl-monogalactosyl-diglyceride (LMG), and 12 iron-sulfur clusters (SF4) (Supplementary Table 2). IsiAs contain a total of 270 Chls which are more abundant in the stromal side than those in the lumenal side, namely, 198 Chls are found in the stromal side and 72 Chls in the lumenal side (Fig. 1c and Supplementary Table 2). The Chls in the stromal side are distributed evenly and form a global network over all IsiAs, whereas Chls in the lumenal side are distributed radially and form distinct clusters within each IsiA monomer (Fig. 1c). This illustrates that Chls in IsiA have less interactions between adjacent IsiA subunits at the lumenal side. IsiAs also contain 72 Bcrs, which surround PSI core and are distributed into two layers, namely, the inner layer and the outer layer (Fig. 1d).

The overall structure and pigment organization of T_PSI–IsiA are similar to those of S_PSI–IsiA[35]. However, compared with the S_PSI–IsiA structure, the T_PSI–IsiA structure has an additional PsaX subunit in the PSI core and there are some differences in the positions of several Chls associated with the PsaX subunit or located near the PsaX position between T_PSI–IsiA and S_PSI–IsiA (Supplementary Fig. 5a, b). Chl101 is coordinated by N23 of PsaX in the T_PSI–IsiA, whereas its equivalent, Chl1302, is coordinated by D56 of PsaF in the S_PSI–IsiA (Supplementary Fig. 5b), and the position of Chl101 is shifted by 8.2 Å toward the stromal side relative to Chl1302 (in terms of the Mg–Mg distance). The loop from residue 305 to 316 in the PsaB subunit interacts with the N-terminal loop of IsiA5 in the T_PSI–IsiA; however, the loop from residue 306 to 320 in PsaB is directed toward inside of the PSI core and does not interact with any IsiAs in the S_PSI–IsiA. Chl1240 of PsaB in the S_PSI–IsiA overlaps with PsaX in the T_PSI–IsiA, and thus has a shifted position designated as Chl834 in the T_PSI–IsiA. While Chl1240 forms a triplet Chl cluster with Chl1218 and Chl1219 in the stromal side in S_PSI–IisA (Supplementary Fig. 5c), Chl841 in the T_PSI–IsiA forms a triplet Chl cluster with Chl834 and Chl840 in the luminal side. These differences may cause differences in the energy transfer behavior between the T_PSI–IsiA and S_PSI–IsiA.

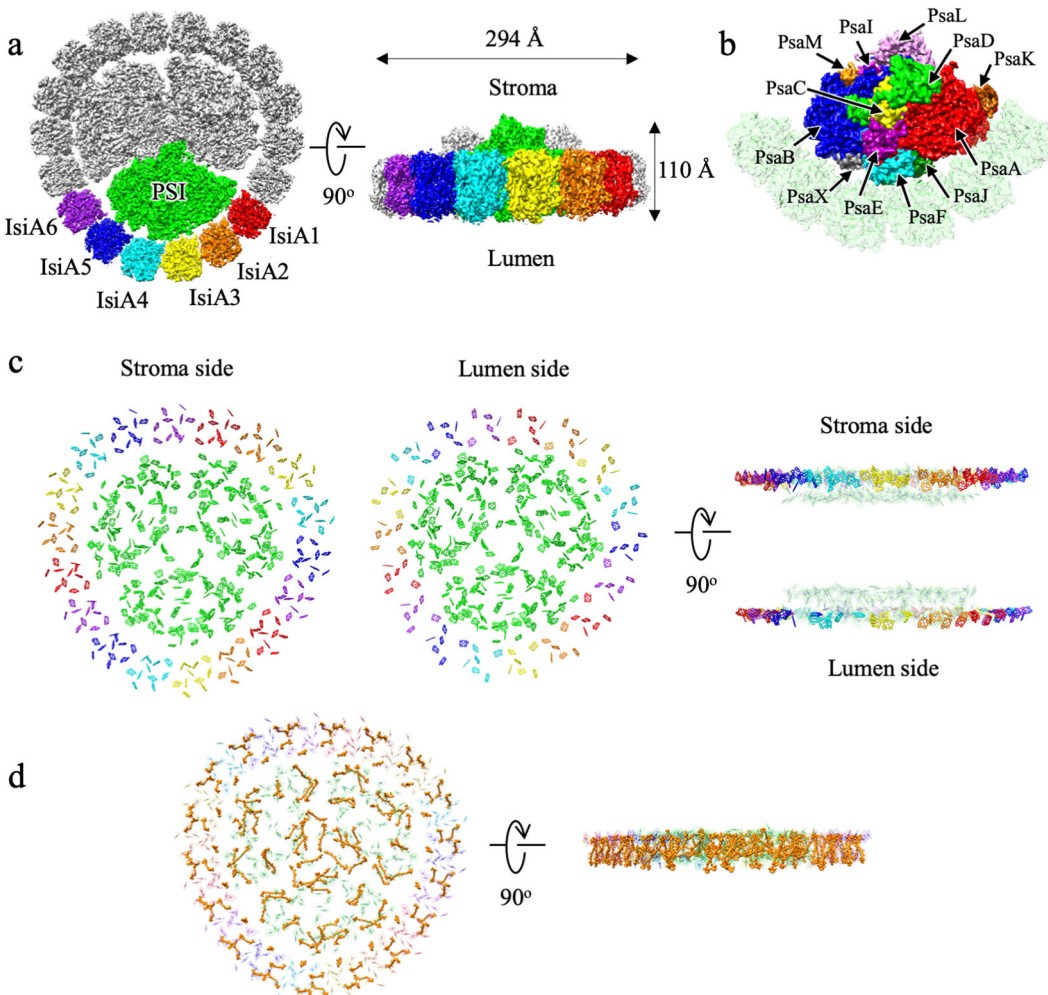

**Fig. 1 Overall structure of the PSI–IsiA supercomplex.** Overall structure of the PSI–IsiA supercomplex was fitted into the cryo-EM map at 2.7 Å resolution.
**a** Left-side: top-view of the cryo-EM map from the stromal side. One of the PSI cores is colored in green, IsiA1–6 are colored in red, orange, yellow, cyan, blue, and purple, respectively, where IsiAs were numbered clockwise. Right-side: side-view of PSI–IsiA. **b** Cryo-EM map for each subunit of the PSI core. Color codes used are red, PsaA; blue, PsaB; yellow, PsaC; green, PsaD; purple, PsaE; cyan, PsaF; magenta, PsaI; dark green, PsaJ; brown, PsaK; pink, PsaL; orange, PsaM; gray, PsaX. **c** Distribution of chlorophylls in PSI–IsiA. Left- and center-panels: top views from the stromal and lumenal sides, respectively. Chls in the PSI core are colored in green, and those in IsiA1–6 are colored in red, orange, yellow, cyan, blue, and purple, respectively. Right panels: side views of the stromal side and lumenal side layers, respectively. **d** Distribution of carotenoids in PSI–IsiA. Left-side is the top-view from the stromal side and right-side is the side-view. Carotenoids are colored in orange.

**Structures of the PSI core trimer and IsiA.** Structure of the PSI core trimer in the T_PSI–IsiA was similar to its crystal structure from *Thermosynechococcus elongatus* (*T. elongatus*) solved at 2.5 Å resolution (PDB: 1JB0)[36] with an RMSD value of 0.60 (Supplementary Fig. 5d); this is in agreement with the fact that the amino acid sequences of PSI subunits from *T. vulcanus* are 100% identical to those from *T. elongatus*. Except for PsaK, the 2.74 Å cryo-EM map has enough quality to allow assignment of the amino acid side chains (Supplementary Fig. 4). The side chains of PsaK could not be assigned unambiguously due to the limited local resolutions, and the loop regions of residues number 1–18, 39–54, and 77–83, of PsaK were disordered, but almost all chlorophylls can be unambiguously modeled in the structure. Only one Chl, Chl1601, which is coordinated by R24 of PsaM, was missing in the PSI core of *T. vulcanus* (Supplementary Fig. 5d). A triplet Chl cluster (Chl1231, Chl1232, and Chl1233) and a dimer Chl (Chl1218 and Chl1219) exist in the lumenal and stromal sides in the PSI core of *T. elongatus*, respectively[36]. Similarly, a triplet Chl cluster (Chl834, Chl840, and Chl841) and a dimeric Chl cluster (Chl821 and Chl822) in the PSI core of

*T. vulcanus* were found. In contrast, the triplet Chl cluster in the lumenal side became a dimeric Chl cluster (Chl1231 and Chl1232), and the dimer Chl cluster in the stromal side became a triplet cluster (Chl1218, Chl1219, and Chl1240), in the PSI core of *S.* sp. PCC 6803 (Supplementary Fig. 5c). These results suggest that the energy transfer and dissipation system may be different between the thermophilic species *Thermosynechococcus* and mesophilic species *Synechocystis*.

The structure of the IsiA monomer (length: 358 residues) was built from residues number 20–350, which is composed of six long transmembrane helices (I–VI), five short helices (N, I′, I″, II′, and IV′) and four β-strands (β1, β2, β3, and β4) arranged in a short anti-parallel β-sheet (Fig. 2a). Five loops connecting the transmembrane helices are designated as A to E loops from the N- to C-termini (Fig. 2a). Each IsiA monomer contains 17 Chls and 4 Bcrs (Fig. 2b and Supplementary Table 2). The central magnesium atoms of Chls are mainly coordinated by His residues, although some Chls are coordinated by Gln or Asn residues or by backbone carbonyls (Supplementary Table 3). Furthermore, two Chls, Chl404 and Chl407, have no direct

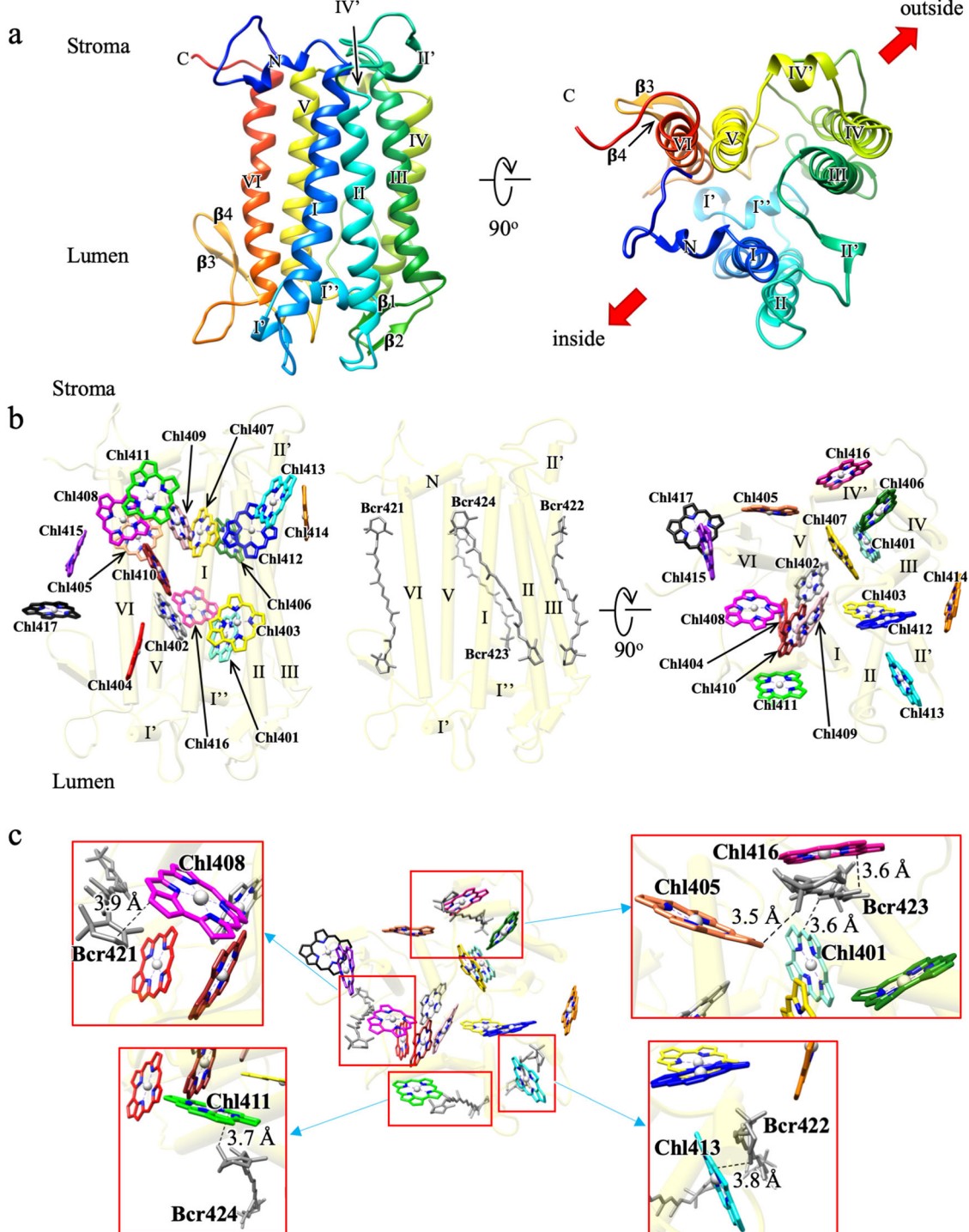

**Fig. 2 Structure of an IsiA monomer and its pigment organization. a** Ribbon diagram of an IsiA monomer. Helices and sheets are classified by Greek number. **b** Arrangement of the pigments (Chls and Bcrs) within the IsiA monomer. Left-side: a side-view represented by stick and light-colored cylinder model. Helices and sheets are classified by Greek number. Chlorophylls are colored by aquamarine (Chl401), gray (Chl402), yellow (Chl403), red (Chl404), coral (Chl405), forest green (Chl406), gold (Chl407), magenta (Chl408), pink (Chl409), brown (Chl410), green (Chl411), blue (Chl412), cyan (Chl413), orange (Chl414), purple (Chl415), deep pink (Chl416), and black (Chl417), respectively. Middle: Arrangement of the Bcrs. Bcrs are colored by gray. Right-side: top-view from the stromal side. **c** Interactions among Chls and Bcrs.

coordinating amino acid ligands, and thus they may be coordinated by water molecules and may have relatively higher excitation-energy levels than other Chls[37].

The structure of IsiA resembles that of PSII CP43 (PDB: 3WU2) as expected by their sequence similarities (Supplementary Fig. 6). However, PSII CP43 has more extended N- and C-terminal residues than IsiA located in the stromal side (1–11 residues in the N-terminal region and 453–473 residues in the C-terminal region) and an additional domain in the lumenal side (305–422 residues). The short additional stromal part of CP43 interacts with neighboring subunits, whereas the large lumenal part maintains the stability of the $Mn_4CaO_5$ cluster and protects it

from attack by outside solute molecules. In addition, the residues coordinating the Chls are well conserved between IsiA and the PSII CP43 proteins (Supplementary Figs. 6 and 7). However, we found four additional Chls, Chl414, Chl415, Chl416, and Chl417 in the IsiA protein, coordinated by H159, Q331, Q206, and carbonyl of I297, respectively (Supplementary Fig. 7 and Supplementary Table 3). These IsiA-unique Chls are conserved between T_PSI–IsiA and S_PSI–IsiA structures (Supplementary Fig. 7 and Supplementary Table 3). Chl414 and Chl416 are located in the peripheral part of the PSI–IsiA complex, and Chl415 and Chl417 are located at the interface between IsiA and the PSI core and hence may be involved in the EET pathways between the IsiAs and PSI core as described below. Some Chls in the IsiA are associated with Bcrs (Fig. 2c), among which, Bcr421 interacts with a Chl and a side chain of PSI core, forming an inner layer within the IsiA ring. The head of Bcr421 in the stromal side interacts with Chl408 at a distance of 3.9 Å (Fig. 2c). On the other hand, the opposite head of Bcr421 in IsiA2, IsiA3, and IsiA4 interacts with Chl817 of PsaA, Chl101 of PsaJ, and Chl102 of PsaJ with distances of 6.3, 4.6, and 5.8 Å, respectively (Supplementary Fig. 8). Therefore, they may mediate EET from IsiA to the PSI core. Bcr422 and Bcr423 contribute to interactions between IsiA subunits, forming the outer Bcr layer. Bcr422 interacts with Chl413 in the stromal side at a distance of 3.8 Å, and Bcr423 interacts with Chl405 in the stromal side at a distance of 3.5 Å, with Chl416 in the middle of the membrane at a distance of 3.6 Å, and with Chl401 in the lumenal side at distance of 3.6 Å, respectively (Fig. 2c). The head of Bcr424 in the stromal side interacts with Chl411 at a distance of 3.7 Å, and may also mediate EET from IsiA to the PSI core (Fig. 2c).

**Interactions and possible EET pathways among different IsiAs.** Each IsiA interacts with their neighboring subunits in a similar manner (Fig. 3). The surface contact areas (buried surface areas) between adjacent IsiA pairs (IsiA1-2, IsiA2-3, IsiA3-4, IsiA4-5, IsiA5-6, and IsiA6-1) are 1395, 1421, 1353, 1412, 1327, and 1152 Å$^2$, respectively (Supplementary Table 4), and the molecular interfaces are formed mainly by hydrophobic interactions via protein–protein, protein–pigment, and pigment–pigment interactions. Hydrophobic patches are observed both at the stromal and lumenal sides. For example, a hydrophobic residue F127 of one IsiA interacts with hydrophobic L338 and F343 residues from the adjacent subunit at the stromal side, and hydrophobic L62 and F105 residues interact with hydrophobic V293 and L295 residues from the neighboring subunit at the lumenal side (Fig. 3c). In addition, five Chls (Chl405, Chl413, Chl415, Chl416, and Chl417) and two Bcrs (Bcr422 and Bcr424) are extensively involved in the inter-subunit interactions between the IsiAs (Fig. 3c, d). Chl405 coordinated by the H333 residue interacts with four hydrophobic residues (I113, V117, A120, and F124) from the neighboring IsiA and also forms a hydrophilic interaction between its carboxy group and R128 of the neighboring IsiA (Fig. 3c, right panel). Chl405 also interacts with Chl413, Bcr422 and Bcr424 from the adjacent subunit (Fig. 3d). Chl417 coordinated by a backbone carbonyl of I297 interacts with hydrophobic residue F60 from the neighboring IsiA (Fig. 3c, right panel). Bcr422 interacts with Chl416 (Fig. 3d), and Bcr424 interacts with the hydrophobic residue L330 and Chls (Chl415 and Chl417) (Fig. 3c and d), from the adjacent subunit, respectively.

To find EET pathways between IsiAs, we calculated the Förster energy transfer rates[3,12,35,38,39] between IsiAs (Supplementary Table 4). Common EET pathways, which are defined by a half-life time of <20 ps, are found between all neighboring IsiA pairs (Fig. 3e). These include three inter-IsiA EET pathways, which

were designated as interior, exterior, and intermediate pathways according to their positions on the IsiA ring relative to the PSI core. All of these pathways are found at the stromal side. In the interior pathway, which is most close to the PSI core, two possible inter-IsiA EET pathways are identified between Chl404 and Chl415, and between Chl410 and Chl417. In the exterior pathway, there are also two possible inter-IsiA EET pathways found between Chl412 and Chl405 and between Chl413 and Chl405. In the intermediate pathway, one possible inter-IsiA EET pathway is found between Chl411 and Chl405. The fastest energy transfer rates between the neighboring IsiAs are always observed between Chl411 and Chl405, and between Chl413 and Chl405, respectively. Thus, these pathways are considered as the main EET pathways between the neighboring IsiAs.

**Interactions between IsiA and PSI.** Interactions of each IsiA with the PSI core are rather diverse (Fig. 4a) which are summarized in Supplementary Table 5. This can be seen in the surface contact areas between the PSI core and the individual IsiA subunits from IsiA1 to IsiA6, which are 267.4, 103.8, 317.0, 371.3, 229.2, and 4.1 Å$^2$, respectively (Supplementary Table 5). Relatively strong interactions with the PSI core are found for IsiA1, IsiA3, isiA4, and IsiA5 (Figs. 4b and d–f and Supplementary Table 5). Especially, the C-terminal region of the IsiA4 is extended and interacts with the PsaF subunit in the PSI core by hydrophobic interactions and thus exhibits the largest surface contact area (Fig. 4e). In contrast, IsiA2 has limited interactions with the PSI core (Fig. 4c), and IsiA6 has almost no contact with the PSI core, since IsiA6 is obviously located far from the PSI core with a distance over 8 Å. These results indicate that IsiA4 might be a key subunit for the assembly of the octadecameric IsiA ring around the PSI core with specific interactions at its C-terminal region. This is apparently different from the S_PSI–IsiA structure[35], where the C-terminus of IsiA contains a flexible amphipathic helix and adopts slightly different orientations at different IsiA positions. The C-terminus of IsiA1–5 in the S_PSI–IsiA is directed to the neighboring IsiA subunit and interacts with the N-terminus of the adjacent monomer, whereas the C-terminus of IsiA6 appears to be disordered. Indeed, the amino acid sequences in the C-terminal region appear less conserved between S_PSI–IsiA and T_PSI–IsiA (Supplementary Fig. 9). Thus, the molecular recognition between PSI and IsiA as well as between different IsiAs may be different between S_PSI–IsiA and T_PSI–IsiA.

**Possible excitation-energy transfer pathways between IsiA and PSI.** The possible EET pathways between IsiA and the PSI core were investigated based on the Förster energy transfer rates among adjacent Chls (Fig. 4h–j and Supplementary Table 5). The following possible EET pathways were identified: 1As, 1Al, 2As, 2Al, 3Al, 3Jl, 4Jl, 5Bl, 5Xl, and 6Bl, which were named according to that used for PSI–LHCI[2,7,40]. As examples of these names, 1As and 1Al represent the pathways identified from IsiA1 to PsaA at the stromal and luminal sides, respectively. The results showed that all IsiAs have substantial EET pathways to the PSI core, among which, the EET pathway from IsiA6 to PSI core is independent of the molecular interactions between this IsiA and PSI core. There are five Chls in IsiAs that are involved in these EET pathways; they are Chl404, Chl408, Chl411, Chl415, and Chl417. In particularly, Chl404 and Chl417 have the fastest energy migration rate between the IsiAs and the PSI core (Supplementary Table 5); therefore these two Chls may play the most important roles for the EET from IsiAs to the PSI core. Chl404 in IsiA5 and Chl841 in PSI had the nearest distance in all EET pathway (Supplementary Table 5). However, Chl841 forms

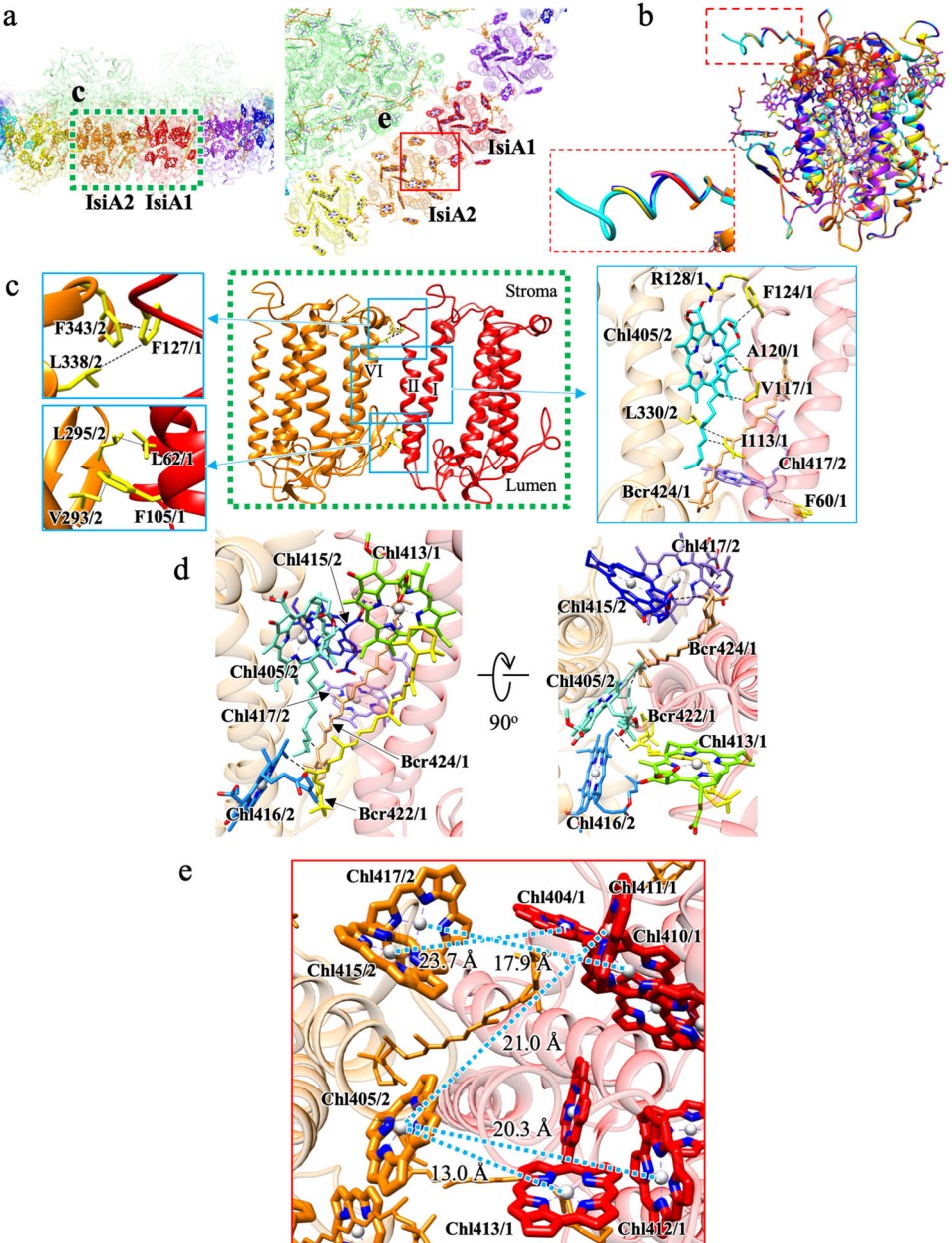

**Fig. 3 Interactions and possible EET pathways within the IsiA ring. a** Overview of interactions between adjacent IsiAs. Squared areas by green dashed and red solid lines are enlarged in panels (**c**) and (**e**), respectively. **b** Superimposition of the structures of the six IsiAs. The squared region by a red dashed line indicates the C-terminal region of IsiA and is enlarged in the left bottom side. The C-terminal region of IsiA4 (cyan) are more ordered than that of other IsiAs. **c** Interactions between IsiA1 and IsiA2. The middle panel is an overview, and the left and right panels show the protein–protein and protein–chlorophylls interactions, respectively. **d** Pigment–pigment interactions between IsiA1 and IsiA2. Right panel is a top-view from the stromal side. **e** Possible EET pathways between IsiA1 and IsiA2.

tripled Chl in PSI, and the pathway might rather work as energy quencher. Among all IsiAs, IsiA1 has three pathways with relatively faster energy migration rates between the IsiAs and the PSI core than those in other IsiAs, suggesting that IsiA1 plays an important role for EET from the IsiA ring to the PSI core.

**Time-resolved fluorescence analyses of PSI–IsiA.** To examine the function of IsiA in *T. vulcanus*, we compared TRF spectra between the PSI–IsiA supercomplex (red lines) and the PSI core trimer (black lines) (Fig. 5a). The spectra of PSI–IsiA and PSI cores exhibited a fluorescence peak at around 730 nm in the time range from 0–4.9 ps to 2.0–2.4 ns. Thus, the peak at around 730 nm was attributed to the fluorescence from Chls with lower

energy in the PSI cores than the energy level of the special pair Chls P700. The existence of the low-energy Chls are confirmed by an absorption peak located at 709 nm (Supplementary Fig. 1). By contrast, a peak at around 683 nm appeared only in the PSI–IsiA spectra particularly at the picosecond time range of 0–4.9 ps after the excitation. Based on its absence in the spectra of the PSI trimer and the blue-shifted Qy absorption spectrum of the PSI–IsiA (Supplementary Fig. 1), the 683-nm peak is attributed to the fluorescence from the IsiAs. The relative intensity of the 683-nm peak decreased gradually, and then almost disappeared in the spectrum in the time range of 180–210 ps. These results indicate that IsiA transfers the excitation energy to the PSI core or quenches the energy within ~180 ps.

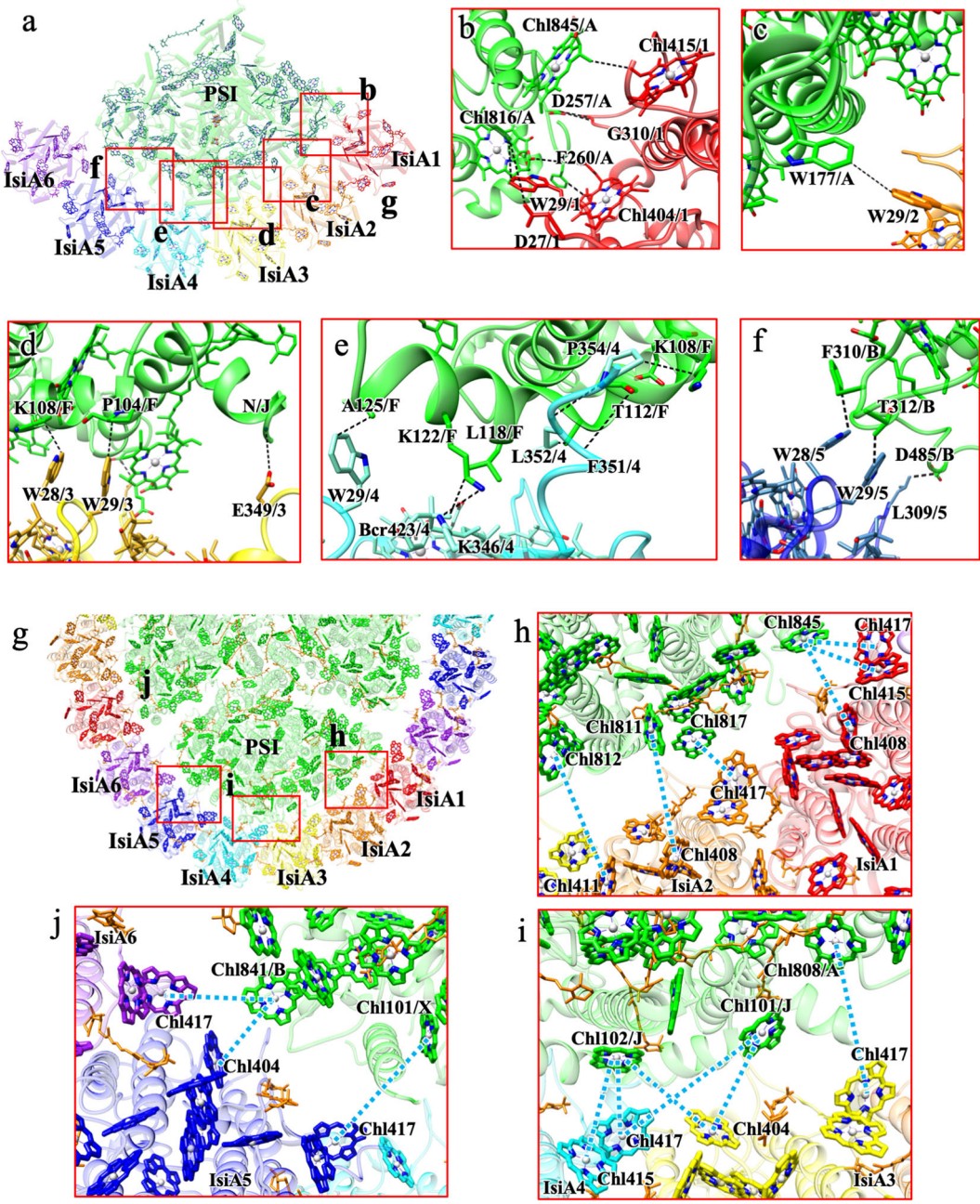

**Fig. 4 Interactions and possible EET pathways between IsiAs and PSI. a** Overview of interactions between IsiAs and PSI. Squared areas are enlarged in panels (**b–f**). **b–f** Interactions between each IsiA and PSI. Interactions were indicated by dashed lines. **g** Overview of possible EET pathways between IsiAs and PSI. Squared areas are enlarged in panels (**h–j**). Possible EET pathways (dashed lines) between each IsiA and PS1.

To examine the detailed excitation-energy dynamics in the PSI–IsiA, we constructed fluorescence decay-associated (FDA) spectra (Fig. 5b). In these spectra, a pair of positive and negative peaks reflects EET from Chls with the positive peak to Chls with the negative peak. The FDS spectrum of PSI–IsiA showed a set of 685-nm positive and 730-nm negative peaks in the time range of 35 ps, indicating EET from IsiA to the PSI core. By contrast, the 50-ps FDA spectrum of the PSI core trimer showed a small positive peak at around 693 nm, indicating that the energy donor in the PSI trimer is different from that in the PSI–IsiA. The FDA spectra of the two samples in the time range of 85–470 ps exhibited only a decay component at around 730 nm, indicating either energy trapping at the RC Chls[41–44] or quenching by Chl–Car interactions[45–48]. The forth FDA spectrum of PSI–IsiA

(3.5 ns) exhibited a positive peak at 685 nm with a very tiny amplitude compared with the peak amplitudes in the first, second, and third FDA spectra (note that the scale of the Y-axis of the forth FDA is enlarged by 1000 times). This indicates that only slight amounts of IsiA appear to be dissociated from the supercomplex. Thus, the FDA analysis strongly indicated functional EET from IsiA to the PSI cores.

To verify whether IsiA is related to energy quenching in the time range of femtoseconds, we measured femtosecond fluorescence decay curves at 685 nm with an upconversion system under room-temperature[49] (Fig. 5c). The fluorescence decay of the PSI–IsiA supercomplex (red line) was apparently slower than that of the PSI trimer (black line), indicating that excitation-energy quenching is not facilitated by the association of IsiA with the PSI

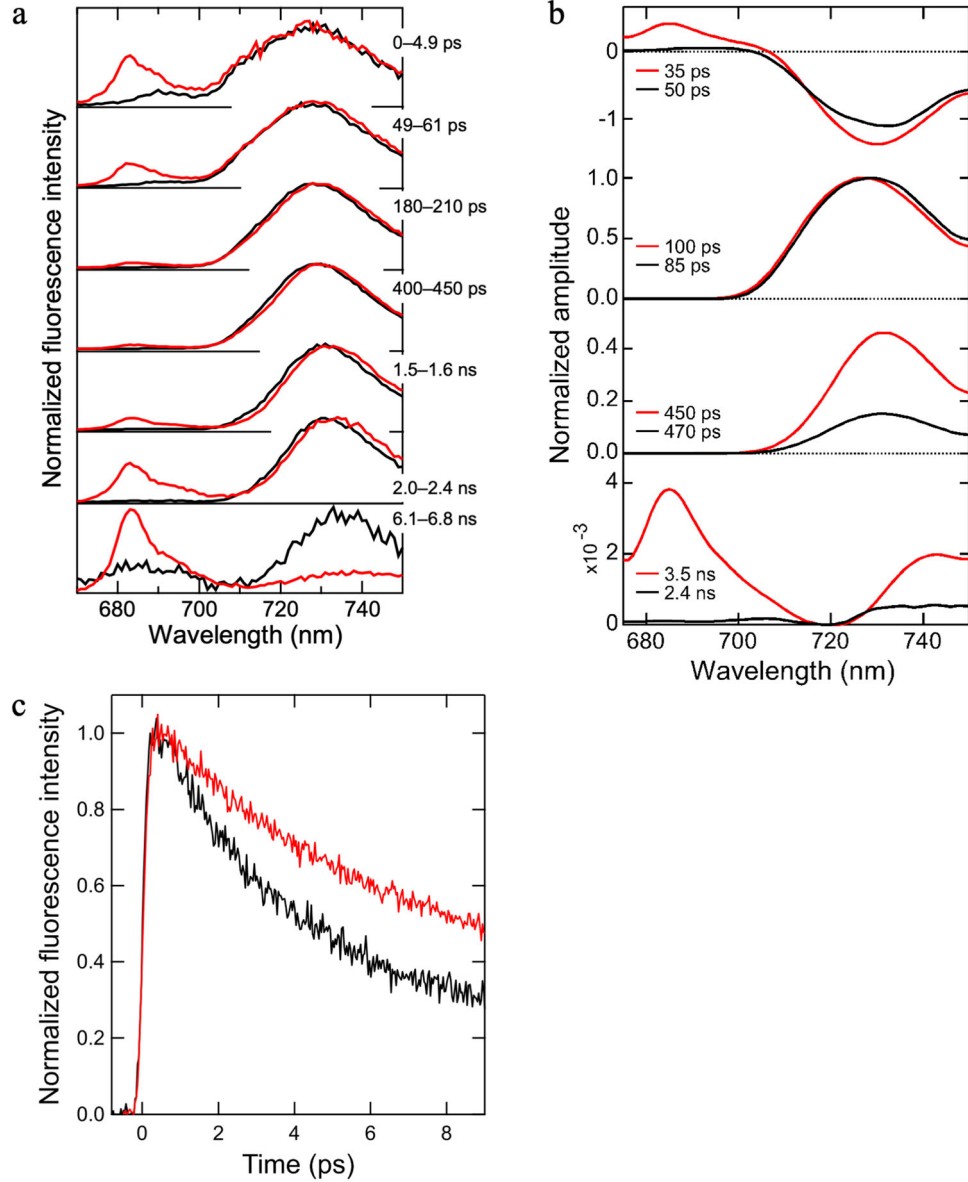

**Fig. 5 TRF analyses of the PSI–IsiA supercomplex. a** 77-K TRF spectra excited at 445 nm. The spectra of the PSI–IsiA and PSI trimer cores were normalized by the maximum intensity of each spectrum. The spectra of the PSI–IsiA and PSI trimer cores are depicted in red and black lines, respectively. **b** 77-K FDA spectra. The spectra of the PSI–IsiA and PSI trimer cores are depicted in red and black lines, respectively. **c** Normalized fluorescence decay curves at 296 K monitored at 685 nm. The curves of the PSI–IsiA and PSI trimer cores are depicted in red and black lines, respectively.

core. Thus, our TRF analyses provides strong evidence that IsiA serves as an energy donor to the PSI core.

## Discussion
The structure and spectroscopic results presented here provide important information regarding the possible EET pathways in the PSI–IsiA supercomplex. Our structure revealed that a trimeric PSI core is encircled by 18 copies of IsiA. All IsiAs are connected with each other in similar manners and form circular EET pathways within the ring surrounding the PSI core trimer. Therefore, the inter-IsiA network may enable the PSI cores in the trimer to share the excitation energies captured by any of the antenna IsiAs efficiently (Fig. 6a). Plausible time constants for the energy migration among the 18 IsiAs should be within 35 ps, because the FDA spectrum of PSI–IsiA showed a time of 35 ps for EET from IsiA to PSI (Fig. 5b). On the other hand, the EET

pathways from IsiAs to the PSI core are apparently different depending on the IsiA positions (IsiA1 to IsiA6). Although all IsiAs have possible EET pathways to PSI, the energy migration rates between the Chls in IsiA1 and the PSI core is relatively faster than those between the other IsiAs and the PSI core (Supplementary Table 5). Thus, IsiA1 has the main EET pathway to PSI between the IsiA ring and the PSI core (Fig. 6a). In addition, two IsiA-unique Chls, Chl415 and Chl417, are extensively involved in the EET pathways between IsiAs and PSI, and both of them are also involved in the inter-IsiA EET pathways. These observations imply that IsiA, probably derived from CP43, acquired these Chls for the efficient EET to PSI. By contrast, other IsiA-unique Chls, Chl414 and Chl416, are located in the peripheral part of the PSI–IsiA complex. As the PSI core is sometimes surrounded by two rings of IsiAs, these Chl414 and Chl416 may face to the possible second, exterior IsiA ring and contribute to EET between the two IsiA rings.

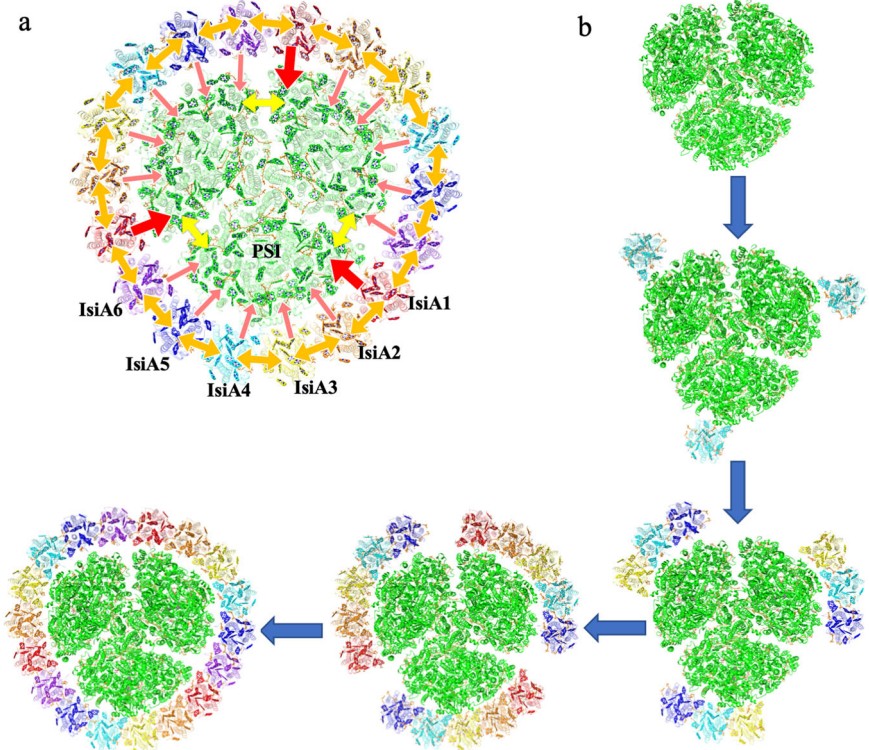

**Fig. 6 Possible EET pathways within PSI–IsiA and a proposed assembly model. a** Overview of possible EET pathways in the whole PSI–IsiA supercomplex. Red arrows indicate EET pathways from IsiA to PSI core; orange arrows indicate EET pathways between adjacent IsiAs; yellow arrows indicate EET pathways between different PSI cores. **b** A proposed assembly model for PSI–IsiA where the assembly of IsiAs starts from the attachment of IsiA4 to the PSI core. See text for more details.

The PSI–IsiA structure obtained here also brings important implications on the assembly mechanism of the PSI–IsiA supercomplex. The 18 IsiAs can be divided into three units, each consisting of six IsiAs, and each of these six IsiAs binds to the PSI core in significantly different manners. This is reflected by the remarkable differences in the contact surface areas between each IsiA and the PSI core. In particular, IsiA4 has the largest contact surface area with the PSI core, and its C-terminal region is in close interaction with the PsaF subunit of the PSI core. Compared with other IsiAs, the C-terminal region of the IsiA4 subunit is well ordered and extended to the PSI core. The strongest interactions between IsiA4 and the PSI core suggest that the IsiA ring formation may be initiated by adsorption of an IsiA to the IsiA4 position on the PSI core, followed by attachment of other IsiAs. On the other hand, IsiAs have an intrinsic ability to form single and double ring assemblies in the absence of the PSI core by their lateral, inter-subunit interactions. This is consistent with the fact that all IsiAs in the PSI–IsiA supercomplex are connected with each other in similar manners. These lines of evidence suggest that the mechanism of IsiA-layer formation is like a traditional epitaxial nucleated-growth type involving an initial nucleation step, in which IsiAs recognize the IsiA4 position, followed by a series of growth steps mainly by lateral inter-IsiA interactions (Fig. 6b). In relation to this, it is interesting to note that the structures of the IsiA C-terminal regions are remarkably different between T_PSI–IsiA and S_PSI–IsiA (Supplementary Fig. 7c). Since the C-terminal region plays an important role in the assembly of the supercomplex, this may suggest a different manner of assembly between T_PSI–IsiA and S_PSI–IsiA.

In summary, this study provides a basis for the possible EET mechanisms in the PSI–IsiA supercomplex of *T. vulcanus* based on its 2.7-Å cryo-EM structure and spectroscopic analyses. The high-resolution structure obtained revealed the tight association

of Chl molecules among neighboring IsiAs within the single ring of the PSI–IsiA supercomplex, enabling a fast energy migration within 35 ps within the IsiA ring, followed by trapping to P700 via Chls positioned at the interface between IsiA and PsaA/PsaB/PsaF. Our femtosecond TRF decay curves do not support the possibility that IsiA serves as an energy quencher, thereby emphasizing the role of IsiA in light-harvesting and energy donation to PSI under iron-stress conditions.

Some structural differences were found between the PSI–IsiA from *T. vulcanus* and *S.* sp. PCC 6803, and these structural differences may bring some functional differences of the IsiAs between the two species. This may be related with the growth environments that each species of the cyanobacterium experiences.

## Methods

**Purification of PSI–IsiA from *T. vulcanus*.** Cells of *T. vulcanus* were cultured in two litters of an iron-free medium at 50 °C for a week to reach to $OD_{720} = 1.0$. The cells were then collected by centrifugation and resuspended with five litters of the iron-free medium, and continued to grow for 2 weeks at 50 °C. Then, $FeCl_3$ was added to a final concentration of 2.5 nM and the culture was further incubated for a week at 50 °C. The cells were pelleted by centrifugation at $13,700 \times g$ for 20 min at 4 °C, and disrupted by lysozyme-treatment and the freeze-thawing method[50,51]. Thylakoid membrane was pelleted by centrifugation at $20,000 \times g$ for 20 min at 4 °C, followed by solubilization with a buffer containing 10 mM $MgCl_2$, 20 mM HEPES–NaOH (pH 7.0), 1.0% (w/v) *n*-dodecyl-*β*-D-maltoside (*β*-DDM) at 4 °C for an hour. Insoluble membrane was removed by centrifugation at $20,000 \times g$ for 10 min at 4 °C. The supernatant was loaded onto a linear trehalose gradient of 10–30% (w/v) in a buffer containing 10 mM $MgCl_2$, 20 mM HEPES–NaOH (pH 7.0), and 0.04% (w/v) *β*-DDM, and centrifuged at $180,000 \times g$ for 4 h at 4 °C. After centrifugation, the band containing PSI–IsiA was collected, and then pelleted with PEG 1500 (at a final concentration of 13% w/v). The pellet was resuspended to 10 mM $MgCl_2$, 20 mM HEPES–NaOH (pH 7.0), and further purified by a 15–30% (w/v) trehalose density gradient centrifugation at $180,000 \times g$ for 20 h at 4 °C. The PSI–IsiA containing band was collected and applied to cryo-EM grids for data set A. For the cryo-EM data set B, trehalose in the sample solution was removed by

pelleting the PSI–IsiA containing band with PEG 1500 (at a final concentration of 13% w/v), and resuspended to a buffer of 50 mM HEPES–NaOH (pH 7.0), 0.04% $\beta$-DDM before being applied to the cryo-EM grids.

**Cryo-EM data collection**. Two image data sets (data set A and B) were acquired. The data set A contains 4902 images and was acquired from holey carbon grids covered with amorphous carbon film, and the data set B contains 15,897 images and acquired from holey carbon grids without amorphous carbon film. For cryo-EM experiments of data set A, a 3-µL aliquot of the PSI–IsiA sample (30 µg of Chl mL$^{-1}$) in a buffer containing 50 mM HEPES–NaOH (pH 7.0), 10 mM MgCl$_2$, 0.04% $\beta$-DDM, 30% trehalose was applied to glow-discharged holey carbon grids (Quantifoil R1.2/1.3, Mo 200 mesh) covered with 5–10 nm amorphous carbon film. The grids were incubated for 30 s in the chamber of FEI Vitrobot Mark IV at 4 °C and 100% humidity, and then washed once with 3 µL of a wash buffer containing 50 mM HEPES–NaOH (pH 7.0) and 0.04% $\beta$-DDM without trehalose. This wash process removes treharose dramatically and enhanced the image contrast of particles as described previously[7]. The washed grids were immediately blotted with filter papers for 3 s and plunged into liquid ethane cooled by liquid nitrogen and then transferred into a cryo-electron microscope (Titan Krios, Thermo Fischer Scientific) equipped with a field emission gun, a Cs corrector (CEOS GmbH), and a direct electron detection camera (Falcon 3EC, Thermo Fischer Scientific) and operated at 300 kV. Image movies were recorded using the Falcon 3EC in a linear mode with a nominal magnification of ×59,000, which results in a final pixel size of 1.113 Å. Each exposure of 2.5 s was dose-fractionated into 32 movie frames, leading to a total electron dose of 50 electrons Å$^{-2}$, with a nominal defocus range of −2.0 to −4.0 µm. For cryo-EM experiments of data set B, a 2.5-µL aliquot of the PSI–IsiA sample (103 µg of Chl mL$^{-1}$, a higher concentration than that used for the sample for the data set A) in a buffer containing 20 mM HEPES–NaOH (pH 7.0), 10 mM MgCl$_2$, and 0.04% $\beta$-DDM was applied to glow-discharged holey carbon grids (Quantifoil R1.2/1.3, Mo 300 mesh). The grids were incubated for 30 s in the chamber of FEI Vitrobot Mark IV at 4 °C and 100% humidity. The grids were blotted with a filter paper manually and another 2.5-µL aliquot of the PSI–IsiA (103 µg of Chl mL$^{-1}$) was applied to the grids in order to increase the number of the particles in the holes. Then, the grids were immediately plunged into liquid ethane cooled by liquid nitrogen. Movies for the data set B were recorded in the same conditions as that for the data set A. All of the image data sets (total 20,799 micrographs) were finally combined and used for the structure analysis.

**Cryo-EM image processing**. Movie frames were aligned and summed using the MotionCor2 software[52] to obtain a final dose weighted image. Estimation of the contrast transfer function (CTF) was performed using the CTFFIND4 program[53]. All of the following processes were performed using RELION3.0[54]. For structural analyses of PSI–isiA, 1,391,531 and 3,109,982 particles were automatically picked from 4902 and 15,897 micrographs in data sets A and B, respectively, and then were used for reference-free 2D classification. Then, 221,114 and 2,623,764 particles were selected from each good 2D classes. Top views (particle view along the membrane normal) were abundant in the data set A, and side views (particle view perpendicular to the membrane normal) were abundant in the data set B (Supplementary Fig. 2a and 2b). These images were merged and subjected to 3D classification with a C3 symmetry. The initial model used for the first 3D classification was generated de novo from particle images in good classes by 2D classification. As described in Supplementary Fig. 2c, 303,983 particles were further refined based on the per-particle defocus and the beam tilt by CTF refinement and Bayesian polishing (Supplementary Fig. 3a). The map was refined by a post-refinement procedure and reconstructed to 2.74 Å resolution (Supplementary Fig. 3a). The resolution was estimated by the golden FSC standard with a 0.143 cutoff[55]. Local resolution was estimated using RELION3.0 (Supplementary Fig. 3c).

**Atomic model building and refinement**. The 2.74 Å map was used for atomic model building of the PSI–IsiA supercomplex. For model building of the PSI core, the crystal structure of *Thermosynechococcus elongatus* PSI (PDB: 1JB0)[36] was first manually fitted into the 2.74 Å cryo-EM map with UCSF Chimera[56], and then adjusted with COOT[57]. For model building of the IsiA subunit, a homology model constructed using the Phyre2 server[58] was first manually fitted into the 2.74 Å cryo-EM map with UCSF Chimera, and then adjusted with COOT. The PSI–IsiA structure was then refined with phenix real-space refinement[59] with geometric restraints for the protein–cofactor coordinations. The final model was further validated with MolProbity[60] and EMringer[61]. The statistics for data collection and structure refinement are summarized in Supplementary Table 1. Figures are made by UCSF Chimera[56]. Surface contact areas were calculated with Areaimol of CCP4 package[62]. The radius of the prove solvent molecule used was 1.4 Å.

**Spectroscopic analysis**. TRF spectra were recorded at 77 K by a time-correlated single-photon counting system with a wavelength interval of 1 nm/channel and a time interval of 2.44 ps/channel. The excitation source used was a picosecond pulse diode laser (PiL044X; Advanced Laser Diode Systems, Germany) operated at 445 nm with a repetition rate of 3 MHz. The FDA spectra were constructed by global analysis according to the previous method[41], but using Mathematica (Wolfram Research). Fluorescence rise and decay curves were measured with a femtosecond

fluorescence upconversion method[49]. The excitation pulse was the second harmonic (425 nm) of a Ti:sapphire laser (Tsunami, Spectra-Physics, USA) (80 MHz) pumped with a diode-pumped solid-state laser (Millennia Xs, Spectra-Physics, USA). To avoid polarization effects, the angle between the polarizations of the excitation and probe beams was set to the magic angle by a $\lambda/2$ plate. The instrumental response function had a pulse width of ~200 fs. Absorption spectra at 77 K were measured by a spectrometer equipped with an integrating sphere unit (V-650/ISVC-747, JASCO, Japan).

**Förster energy transfer rate calculation**. Förster energy transfer rates were calculated according to the Förster theory[3,12,35,38,39]. The Förster energy transfer rate $K_{DA}$ is defined as $K_{DA} = (C_{AA}\kappa^2)/(n^4 R_{DA}^6)$, where $C_{AA}$ is a factor calculated from the overlap integral between Chls, $\kappa^2$ is the dipole orientation factor, $n$ is the refractive index and $R_{DA}$ is the distance between magnesium of Chls. The $C_{AA}$ value for Chl $a$ to Chl $a$ energy transfer and the $n$ value were 32.26 and 1.55, respectively, as estimated by Gradinaru et al.[39]. The dipole orientation factor $\kappa^2$ is defined as $\kappa^2 = (\cos\alpha - 3\cos\beta_1\cos\beta_2)^2$, where $\alpha$ is the angle between the two transition dipoles and $\beta$s are the angles between each dipole and the Mg–Mg line[63,64].

**Statistics and reproducibility**. No statistical method was used to predetermine the sample size, and the experiments were not randomized. The investigators were not blinded to allocation during experiments and outcome assessment. The purification of PSI–IsiA was repeated over three times, which showed same results. The spectroscopic analysis was performed at least two times, and similar results were obtained. The cryo-EM data was collected from four grids. Individual images with bad ice were excluded from the data set by visual inspection. Data collection, processing and refinement statistics were summarized in Supplementary Table 1.

**Reporting summary**. Further information on research design is available in the Nature Research Reporting Summary linked to this article.

## Data availability
Atomic coordinates and Cyro-EM maps for the reported structure of PSI–IsiA have been deposited in the Protein Data Bank under an accession code 6K33 and in the Electron Microscopy Data Bank under an accession code EMD-9908. Source data of Fig. 5 is provided as Supplementary Data 1. Remaining data supporting this study are available from the corresponding author upon reasonable request.

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

## Acknowledgements

This work was supported by the Platform Project for Supporting Drug Discovery and Life Science Research (Basis for Supporting Innovative Drug Discovery and Life Science Research (BINDS)) from AMED under Grant Number JP18am0101072j0001 (to N.M.), PRESTO from JST Grant No. JPMJPR16P1 (to F.A.), JSPS KAKENHI No. JP17K07442 and JP19H04726 (to R.N.), JP16H06553 (to S.A.), and JP17H06434 (to J.-R.S.).

## Author contributions

F.A. and N.M. conceived the project; F.A. and Y.N. purified the PSI–IsiA supercomplexes; R.N. characterized biochemical features; R.N., M.Y., Y.U., and S.A. measured time-resolved fluorescence spectra and performed data analyses; T.S. and N.D. identified gene products of IsiA by MS analyses; N.M. collected cryo-EM images; K.K., F.A., and N.M. processed the EM data and N.M. reconstructed the final EM maps; F.A., K.K., and N.M. built the structure model; K.K. refined the final models; F.A. analyzed the structure; F.A., R.N., K.K., Y.N., J.-R.S., S.A., and N.M. wrote the paper, and all of the authors joined the discussion of the results.

## Competing interests

The authors declare no competing interests.
