## [Peer Review File · Communications Biology]

Reviewers' comments:

Reviewer #1 (Remarks to the Author):

1. The triplet Chl (B33-B32-B31 in PDB:1JB0) was partially lost in the PSI of *Synechocystis* sp. PCC6803. This is one of the important difference between *Synechocystis* and *T. vulcanus*. Please discuss the energy transfer between *isiA* and triplet Chl in PSI. The location of this triplet Chl was apart from electron transfer chain. It may work as energy quencher. Is there a possibility of the presence of energy transfer from triplet Chl to *isiA*?
2. LH1 and LH2 in purple photosynthetic bacteria also formed circle structure. Their excitation energy were shared among each subunit. Is it possible to share energy among *isiA* subunits in the view of the structure?
3. Is it possible to overlay the ring of *isiA* in *Synechocystis* sp. PCC6803 on your *isiA*? How about the effect of *PsaX* in the structure of *isiA*?
4. The authors used four time parameter to calculate FDA spectrum. Why the authors selected the four components? To discuss the peak maximum of TRF, the absorption spectra of PSI and *isiA*-PSI are also necessary.

Minor

Several typos were found your MS.

Line 64: donation

line 121: supercomplex

line 178: extensively

line 477: side

Reviewer #2 (Remarks to the Author):

The manuscript by Akita et al describes the cryo-EM structure of the photosystem I trimer associated with 18 *IsiA* subunits from *Thermosynechococcus vulcanus*. A similar structure was published a few months ago from the cyanobacteria *Synechocystis* 6803, however, given the differences between the two strains the current structure is of significant scientific interest and merits publication in *Communications biology*.

The reported resolution of around 2.7Å is high and should be enough to obtain significantly better refinement statistics. I am especially referring to the high number deviations from ideal values for many protein residues, which are higher than expected at the reported resolution. The authors must show the FSC for their phase randomized volumes to validate their final mask and it would be better if they provide FSC of map and model in addition to the curve they show on supplementary figure 3a. Other than that, the quality of experimental work is high and supports the conclusions the authors draw from their map.

One of the major differences between 6803 and *vulcanus* is the presence of *PsaX*. While the authors mention this in the text (lines 115-118), they do not show the associated differences and discuss whether accompanying differences exist in *IsiA*.

Given that this is the first verification of the nearly iconic structure of PSI from thermophilic cyanobacteria and the relatively high resolution, I think a superposition of the current model of the PSI trimer and 1jb0 can be included in the supplementary information, the text only mentions very low rmsd values. If there are truly no differences in pigment positions, then this suggestion can be

omitted.

Throughout the text the author refers to Buried surface area as a measure for interaction. This is fine, but the authors should specify how this was calculated and how different ligands were treated, especially when referring to IsiA-IsiA contacts. What distances were used as cutoff values for interacting surfaces?

Throughout the text the authors refer to a scientific discussion regarding the role of IsiA as a quencher when associated with photosystem I. To my understanding all the literature that refers to IsiA as a quencher of chlorophyll excited state refers to IsiA when it is not associated with PSI. Specifically, the authors mention the previous publication of the structure by Toporik et al as stating that IsiA functions as a quencher when associated with PSI, which is a misrepresentation of the conclusions in this paper. The authors claim that the present structure and the S_PSI-IsiA structures are "remarkably different" (line 319) and point to supp. Figure 6C which shows a superposition of a single IsiA monomer. It is not clear if they superposed IsiA from similar positions in the complex. In addition, the authors point that the C terminal of the T_PSI-IsiA structure is configured differently than the S_PSI-IsiA (line 213). In figure 3B the c terminal helices of at least some of the IsiA subunits appear partially disordered, it would be nice to carry a more systematic comparison, perhaps defining which IsiA positions are most similar between the two structures or update figure 3B.

The authors discuss EET pathways between IsiA and the PSI core. The authors measured fluorescence decay kinetics in PSI trimers and in PSI-IsiA at 77K. I agree that the results show EET from IsiA to PSI, but that was never in question. The authors use the words "portal" and "gateway" when describing IsiA1, even though IsiA1 may be more tightly connected to PSI, using these terms seems erroneous to me, as clearly a substantial amount of EET can occur throughout the ring. I leave this to the discretion of the authors.

Minor points –

The authors refer to LHCs at the beginning of the introduction, without first stating that they refer to a protein family. In my opinion this is somewhat confusing, and readers may assume the authors refer to any light harvesting protein complex.

Line 47 - "different living environments" should be replaced with a more suitable phrase.

Line 58 - "but not with PSII examined so far" - rephrase

Line 65 and 69- references are missing

Line 91- sup table 1 contains data collection statistics and should not be referred to here.

Line 477- "right-side" should be corrected.

Figure 3 : the green square in 'a' is presumably switched to a side orientation in 'c'.

Sup figure 6 : a- the chlorophyll number is covered by figure b. Sup figure 4 and 7 – no description of the features in the figure.

Synechocystis sp. PCC 6803 or S.sp. PCC 6803– choose one.

Sup table 1 – pixel

Sup table 3 – isiA

Reviewer #3 (Remarks to the Author):

This work presents the latest approach to solve IsiA-PSI supercomplex molecular structure with increased resolution. The structural work is accompanied by time-resolved fluorescence studies of the supercomplex.

Authors, with application of modelling of the cryo-electron microscopy images were able to achieve the structure with 2.7 Angstrom resolution that is a substantial improvement compared to 3.5 Angstrom resolution of another IsiA-PSI supercomplex from different bacterial strain published very recently by other group.

This work is interesting and worth publication but there are many, in my opinion, quite problematic

issues, especially in the spectroscopic part that should be clarified/corrected first.

Major issues:

Lines 63-67: "There is a long-standing question as to whether the PSI-specifically bounded IsiA is involved in either excitation-energy donation to PSI or energy quenching by interactions of Chls with carotenoids. Although various spectroscopic analyses were attempted to address this question, a consensus for the function of IsiA has not been reached so far."

This is a strong statement but it is not supported with even one reference by authors. There are no references provided after the sentence....so it really correct?

According to my knowledge there was/is quite good consensus that IsiA is a very good energy donor to PSI (Andrizhiyevskaya et al, *Biochim. Biophys. Acta* 1556 (2002) 265–272.; Melkozernov et al, *Biochemistry* 42 (2003) 3893–3903; Chen et al, *BBA-Bioenergetics* (2017), 1858, 249-258) and there was never under question if IsiA serves as a PSI quencher as authors suggest in this work (it does not). This is my understanding of "quenching" role of IsiA that authors introduced here (it is not really clear from the statement but it reveals that way later in the text where TRF is introduced).

However...IsiA is commonly observed to form structures without PSI and those were identified to have substantially shortened Chl a fluorescence. Because of that question was asked about mechanism of intra-IsiA excitation quenching that has nothing to do with "hypothetical" quenching of PSI by IsiA and involvement of carotenoids in it, as suggested by authors. (Ihalainen et al, *Biochemistry* 44 (2005) 10846–10853; Berera et al, *Chem. Phys.* 373 (2010) 65–70; Berera et al, *Biophys. J.* 96 (2009) 2261–2267; Chen et al, *BBA-Bioenergetics* (2017), 1858, 249-258)

If authors meant IsiA as PSI quencher I would like to ask to elaborate this issue more. Why do authors think it is feasible and what do authors expect to observe in context of TRF measurements as performed here?

Line 156-157: "The head of Bcr421 in the stromal side interacts with Chl408 at distance of 3.9 Å, and therefore may mediate EET from IsiA to the PSI core."

My simple question here is: How? What does author think by saying "mediate EET"? Do they mean to be physically involved in excitation energy transfer process between IsiA and PSI so we have Chl (IsiA)-Bcr(IsiA)-Chl(PSI) pathway for excitation energy transfer? Or authors are trying to "furnish" this pigment into hypothetical quenching mechanism of PSI by IsiA? If it is correct, this is very, very unlikely to happen (for a multiple reasons).

Line 165-221: I think those two paragraphs are exaggerated. All what authors have are pigment-pigment or pigment-protein residue distances based on the molecular structure. Distances are not equal to inter-pigment interactions (the way how pigment molecules interact on each other via energetic coupling, etc.) that could be calculated based on pigment distances and their mutual orientations (electric dipoles). It is rather pigment-ligand framework that shows places where possible, important pigment-pigment interactions may (but not necessary) occur. In this context Figure 4 is pretty much useless as it does not explain anything.

Lines: 240 - 274, I have hard time to follow authors understanding that stands behind performing TRF measurements in this way and their interpretation. Why would authors do fluorescence decay measurements at 77 K for PSI and IsiA-PSI but then compare fluorescence dynamics at 685 nm from room temperature TRF data that are not even shown at all? I thought that PSI has no detectable fluorescence at room temperature so how come it is detectable here. It makes little sense for me. Just assuming (I guess) that authors tried to somehow reveal reverse excitation energy transfer from PSI to IsiA (in other words PSI quenching by IsiA) it would be most useful to excite PSI Qy band just

past 680 nm that minimally overlaps with Qy absorption band of the IsiA and make sure that electrochemistry of the special pair in the PSI is disrupted so potential mechanisms of overexcitation mitigation, like excitation quenching, would trigger. All what is provided in materials and methods is "The light source was a Ti:Sapphire laser (Tsunami, Spectra-Physics, USA)." which is very vague statement without important details. Tsunami oscillator has tuning range between 700 and 1080 nm and it cannot fit in any way to absorption spectrum of IsiA/PSII to be used as excitation source. Provide more details.

Minor points:

- Improve language. There are apparent mistakes in many places: e.g. – line 44 "...harvesting solar energy and transferring them....", line 64 "donoation", line 178 "extenisvly", etc.
- Line 72, 101, and so on: IsiA subunits – IsiA is not a subunit of something nor has any subunits itself (like PsaA, B,... in PSI). It is just a protein complex.
- Line 137, and on vs Figure 2 caption: In the main text and in Figure 2 authors use ChIXXX notation while in the Figure 2 caption CLXXX is used. Decide which one you would like to use.
- Line 250: "...the IsiA molecule" – IsiA is not a molecule. Molecule is group of atoms chemically bound. As IsiA consists of protein and pigments that are not chemically bond so IsiA is not a molecule.
- Figure 2b: it is very congested and will be extremely difficult to see anything on it in the printed version
- Figure 3 d and c: not the best choice of colors to mark Chls and Bcrs. It is extremely difficult to see what is what as Bcrs and Chls are in the same color.
- Figure 5: add legends to a-c to improve figure clarity (black –PSI and red-IsiA-PSI)

Dariusz M. Niedzwiedzki, Ph.D.

Our responses to reviewers' comments are as follows:

Responses to Reviewer #1

Comment 1:

1. The triplet Chl (B33-B32-B31 in PDB:1JB0) was partially lost in the PSI of *Synechocystis* sp. PCC6803. This is one of the important difference between *Synechocystis* and *T. vulcanus*. Please discuss the energy transfer between isiA and triplet Chl in PSI. The location of this triplet Chl was apart from electron transfer chain. It may works energy quencher. Is there a possibility the presence of energy transfer from triplet Chl to isiA?

Author reply 1-1:

Thanks for your comments. You are right that the triplet Chl (B33-B32-B31 in PDB: 1JB0) was partially lost in PSI of *Synechocystis* sp. PCC6803. Actually, this triplet Chl cluster becomes a dimeric Chl cluster in S_6803 PSI, but the same triplet Chls were found in our *T. vulcanus* structure. This triplet Chl cluster is located at the luminal side. However, while a dimeric Chl pair is found in the stromal side of the two thermophilic cyanobacterial PSI structures, a triplet Chl cluster existed in the corresponding position of S_6803 PSI in the stromal side. This may suggest differences in the energy transfer pathways between the thermophilic and mesophilic PSIs. We added the differences between T_PSI-IsiA and S_PSI-IsiA to supplementary Figure 5c, and discussed the differences and their functional implications regarding the triplet and dimeric Chls between T_PSI-IsiA and S_PSI-IsiA on lines 132 to 137 and lines 150 to 159.

Comment 2:

2. LH1 and LH2 in purple photosynthetic bacteria also formed circle structure. Their excitation energy were shared among each subunit. Is it possible to share energy among isiA subunits in the view of the structure?

Author reply 1-2:

Yes, it is possible for the IsiA subunits to share the excitation energy within the IsiA-ring. We added this possibility and discussed them on lines 221 to 234. Our structure suggests that Chl405, Chl411 and Chl413 in IsiAs are important for energy transfer between neighboring IsiAs.

Comment 3:

3. Is it possible to overlay the ring of isiA in *Synechocystis* sp. PCC6803 on your isiA? How about the effect of PsaX in the structure of isiA?

Author reply 1-3:

According to your comment, we added the superposition of T_PSI-IsiA and S_PSI-IsiA to supplementary Fig. 5, and added the sentences regarding differences around the region of PsaX between T_PSI-IsiA and S_PSI-IsiA on lines 121 to 137. PsaX did not interact with IsiA directly, therefore, we consider that PsaX does not affect the structure of IsiA in the PSI-IsiA supercomplex. The Chl associated with PsaX, Chl101, is one of the Chl related to EET pathways between IsiA5 to PSI. Therefore, we consider that PsaX works as an energy acceptor from IsiA to PSI.

Comment 4:

4. The authors used four time parameter to calculate FDA spectrum. Why the authors selected the four components? To discuss the peak maximum of TRF, the absorption spectra of PSI and isiA-PSI are also necessary.

Author reply 1-4:

Our previous TRF studies of PSI cores and PSI-antenna supercomplexes have shown that their FDA spectra can be fitted by four components: for example, those from *Synechocystis* sp. PCC 6803 (Photochem. Photobiol. 86, 62 (2010)) and diatoms that are eukaryotic algae (J. Phys. Chem. B 123, 66 (2019)). These results strongly indicate that the four components are a good fit for the FDA spectra irrespective of prokaryotes and eukaryotes, as well as for both PSI core and PSI-antenna supercomplex. Therefore, we selected the four components for the FDA spectra in the present study.

According to your comments, we added the absorption spectra (shown below) into Supplementary Fig. 1, and modified the results and methods sections accordingly.

Lines 285-286: “The existence of the low-energy Chls are confirmed by an absorption peak located at 709 nm (Supplementary Fig. 1).” was added.

Lines 288-289: “Based on its absence in the spectra of the PSI trimer and the blue-shifted Qy absorption spectrum of the PSI-IsiA (Supplementary Fig. 1),” was inserted.

Line 498-500: “Absorption spectra at 77 K were measured by a spectrometer equipped with an integrating sphere unit (V-650/ISVC-747, JASCO, Japan).” was added.

Minor

Several typos were found your MS.

Line 64: donoation

line 121: supercompolex

line 178: extenisively

line 477: side

Author reply 1-5:

We modified all of them; thank you very much.

Responses to Reviewer #2

The manuscript by Akita et al describes the cryo-EM structure of the photosystem I trimer associated with 18 IsiA subunits from *Thermosynechococcus vulcanus*. A similar structure was published a few months ago from the cyanobacteria *Synechocystis* 6803, however, given the differences between the two strains the current structure is of significant scientific interest and merits publication in *Communications biology*.

The reported resolution of around 2.7Å is high and should be enough to obtain significantly better refinement statistics. I am especially referring to the high number deviations from ideal values for many protein residues, which are higher than expected at the reported resolution. The authors must show the FSC for their phase randomized volumes to validate their final mask and it would be better if they provide FSC of map and model in addition to the curve they show on supplementary figure 3a. Other than that, the quality of experimental work is high and supports the conclusions the authors draw from their map.

Author reply 2-1:

Thanks for your positive and encouraging comments. According to your comments, we added the FSC of map and model to supplementary Fig. 3a. We also added FSC of corrected maps, unmasked maps, masked maps and phase randomized masked maps to supplementary Fig. 3a.

One of the major differences between 6803 and *vulcanus* is the presence of PsaX. While the authors mention this in the text (lines 115-118), they do not show the associated differences and discuss whether accompanying differences exist in IsiA.

Author reply 2-2:

According to your comments, we added a superposition figure between T_PSI-IsiA and S_PSI-IsiA focused on PsaX as supplementary Fig. 5b, and added sentences to describe the differences in the region around PsaX between T_PSI-IsiA and S_PSI-IsiA on lines 121 to 137. "PsaX does not interact with IsiA; however, Chl101 associated with PsaX is close to Chl417 of IsiA5 which may therefore provide one of the EET pathways between IsiA to PSI."

Given that this is the first verification of the nearly iconic structure of PSI from thermophilic cyanobacteria and the relatively high resolution, I think a super position of

the current model of the PSI trimer and 1jb0 can be included in the supplementary information, the text only mentions very low rmsd values. If there are truly no differences in pigment positions, then this suggestion can be omitted.

Author reply 2-3:

Thanks for your comments. According to your comment, we added the superposition of the structures of PSI from *T. elongatus* (1JB0) and *T. vulcanus* (present study) as supplementary Fig. 5d. The rmsd value was very small (0.60), and the only significant difference between the two structures is the absence of Chl1601 coordinated by PsaM in the *T. vulcanus* structure. We described this difference in the text (lines 140-154).

Throughout the text the author refers to Buried surface area as a measure for interaction. This is fine, but the authors should specify how this was calculated and how different ligands were treated, especially when referring to IsiA-IsiA contacts. What distances were used as cutoff values for interacting surfaces?

Author reply 2-4:

We calculated the buried surface areas with Areaimol in the CCP4 package. The radius of probe solvent molecule used was 1.4 Å. These were added to the Methods section on lines 483 to 484.

Throughout the text the authors refer to a scientific discussion regarding the role of IsiA as a quencher when associated with photosystem I. To my understanding all the literature that refers to IsiA as a quencher of chlorophyll excited state refers to IsiA when it is not associated with PSI. Specifically, the authors mention the previous publication of the structure by Toporik et al as stating that IsiA functions as a quencher when associated with PSI, which is a misrepresentation of the conclusions in this paper.

Author reply 2-5:

Thanks for your comments. According to your comments, we modified the text regarding the role of IsiA as a quencher to indicate that this happens only for free IsiAs. In fact, our time-resolved results did not support the role of quenching for IsiAs when it is associated with PSI. We have modified our discussions and conclusions to make this clear.

The authors claim that the present structure and the S_PSI-IsiA structures are

“remarkably different” (line 319) and point to supp. Figure 6C which shows a superposition of a single IsiA monomer. It is not clear if they superposed IsiA from similar positions in the complex. In addition, the authors point that the C terminal of the T_PSI-IsiA structure is configured differently than the S_PSI-IsiA (line 213). In figure 3B the c terminal helices of at least some of the IsiA subunits appear partially disordered, it would be nice to carry a more systematic comparison, perhaps defining which IsiA positions are most similar between the two structures or update figure 3B.

Author reply 2-6:

Yes, we used IsiAs from similar positions of T_PSI-IsiA and S_PSI-IsiA to compare their structures, which is IsiA4 in T_PSI-IsiA. According to your comments, we added an enlarged figure of the C-terminals of six IsiAs to figure 3b to show their differences. The overall structures of T_PSI-IsiA and S_PSI-IsiA were compared in Supplementary Fig. 5a.

The authors discuss EET pathways between IsiA and the PSI core. The authors measured fluorescence decay kinetics in PSI trimers and in PSI-IsiA at 77K. I agree that the results show EET from IsiA to PSI, but that was never in question. The authors use the words “portal” and “gateway” when describing IsiA1, even though IsiA1 may be more tightly connected to PSI, using these terms seems erroneous to me, as clearly a substantial amount of EET can occur throughout the ring. I leave this to the discretion of the authors.

Author reply 2-7:

According to your comments, we removed the words “gateway” and “portal” from the sentence on line 331 to 332, and modified the text to indicate that EET can occur throughout the ring to PSI core, although that from IsiA-1 to the PSI core may contribute a larger part.

Minor points –

The authors refer to LHCs at the beginning of the introduction, without first stating that they refer to a protein family. In my opinion this is somewhat confusing, and readers may assume the authors refer to any light harvesting protein complex.

Author reply 2-8:

We modified the first sentence of text to indicate that LHCs are a family of pigment-proteins, according to your comments.

Line 47 - “different living environments” should be replaced with a more suitable phrase.

Author reply 2-9:

Thanks for your comment. We modified the original sentence to “...various LHCs have been developed to capture the solar energy under different light environments.”

Line 58 – “but not with PSII examined so far” - rephrase

Author reply 2-10:

We removed the phrase “examined so far” from the sentence.

Line 65 and 69- references are missing

Author reply 2-11:

We added references; thank you.

Line 91- sup table 1 contains data collection statistics and should not be referred to here.

Author reply 2-12:

We removed it; thank you.

Line 477- “right-sdie” should be corrected.

Author reply 2-13

We corrected it; thank you.

Figure 3 : the green square in ‘a’ is presumably switched to a side orientation in ‘c’.

Author reply 2-14:

Thanks for your comment. We added the side view to Figure 3a.

Sup figure 6 : a- the chlorophyll number is covered by figure b. Sup figure 4 and 7 – no description of the features in the figure.

Author reply 2-15:

Thanks for your comments. We modified the Supp. figure 7a (originally Supp. Fig. 6).

We added the features to the Supp. Figs. 5 and 8 (originally Supp. Figs. 4 and 7).

Synechocystis sp. PCC 6803 or *S.sp.* PCC 6803– choose one.

Author reply 2-16:

We unified it as “*S. sp.* PCC 6803”.

Sup table 1 – pixel

Sup table 3 – isiA

Author reply 2-17:

We modified them; thank you.

Responses to Reviewer #3

This work presents the latest approach to solve IsiA-PSI supercomplex molecular structure with increased resolution. The structural work is accompanied by time-resolved fluorescence studies of the supercomplex. Authors, with application of modelling of the cryo-electron microscopy images were able to achieve the structure with 2.7 Angstrom resolution that is a substantial improvement compared to 3.5 Angstrom resolution of another IsiA-PSI supercomplex from different bacterial strain published very recently by other group. This work is interesting and worth publication but there are many, in my opinion, quite problematic issues, especially in the spectroscopic part that should be clarified/corrected first.

Comment 1:

Major issues:

Lines 63-67: “There is a long-standing question as to whether the PSI-specifically bounded IsiA is involved in either excitation-energy donation to PSI or energy quenching by interactions of Chls with carotenoids. Although various spectroscopic analyses were attempted to address this question, a consensus for the function of IsiA has not been reached so far.”

This is a strong statement but it is not supported with even one reference by authors. There are no references provided after the sentence....so it really correct?

According to my knowledge there was/is quite good consensus that IsiA is a very good energy donor to PSI (Andrizhiyevskaya et al, *Biochim. Biophys. Acta* 1556 (2002) 265–272.; Melkozernov et al, *Biochemistry* 42 (2003) 3893–3903; Chen et al, *BBA-Bioenergetics* (2017), 1858, 249-258) and there was never under question if IsiA serves as a PSI quencher as authors suggest in this work (it does not). This is my understanding of “quenching” role of IsiA that authors introduced here (it is not really clear from the statement but it reveals that way later in the text where TRF is introduced).

However...IsiA is commonly observed to form structures without PSI and those were identified to have substantially shortened Chl a fluorescence. Because of that question was asked about mechanism of intra-IsiA excitation quenching that has nothing to do with “hypothetical” quenching of PSI by IsiA and involvement of carotenoids in it, as suggested by authors. (Ihalainen et al, *Biochemistry* 44 (2005) 10846–10853; Berera et al, *Chem. Phys.* 373 (2010) 65–70; Berera et al, *Biophys. J.* 96 (2009) 2261–2267; Chen et al, *BBA-Bioenergetics* (2017), 1858, 249-258)

If authors meant IsiA as PSI quencher I would like to ask to elaborate this issue more. Why do authors think it is feasible and what do authors expect to observe in context of TRF measurements as performed here?

Author reply 3-1:

First of all, we thank the reviewer for his/her valuable comments to improve our manuscript.

We agree with the reviewer's comments that IsiA serves as an energy donor to PSI instead of energy quenching from PSI when it is associated with PSI. However, previous studies suggested by the reviewer cannot exclude the possibility that the energy quenching by IsiAs in the PSI-IsiA supercomplexes occurs in very early time region like femtoseconds under physiological-temperature conditions.

To address this question, in this study, we measured two types of TRF measurements: the time-correlated single photon counting (TCSPC) method and the fluorescence upconversion (UPC) method. The latter UPC is a powerful method for detecting fluorescence changes in a femtosecond time scale under physiological-temperature conditions. The result clearly showed excitation-energy transfer from IsiA to PSI.

Based on these results and the comments of the reviewer, we dramatically improved our manuscript and cited new references pointed out by the reviewer, as follows:

Line67-72: "IsiA has been reported to play a role in donating energy to the PSI core in the PSI-IsiA supercomplexes^{refs-1}, whereas free IsiA is likely involved in energy quenching once IsiA is detached from PSI^{refs-2}. However, these spectroscopic results cannot exclude the possibility that energy quenching by IsiA may also occur in the PSI-IsiA supercomplex in a very early time region like femtoseconds under physiological-temperature conditions."

Refs-1: "Andrizhiyevskaya et al, *Biochim. Biophys. Acta* 1556 (2002) 265–272.; Andrizhiyevskaya et al, *Biochim. Biophys. Acta* 1656 (2004) 104–113; Melkozernov et al, *Biochemistry* 42 (2003) 3893–3903; Chen et al, *BBA-Bioenergetics* (2017), 1858, 249-258"

Refs-2: "Ihalainen et al, *Biochemistry* 44 (2005) 10846–10853; Berera et al, *Chem. Phys.* 373 (2010) 65–70; Berera et al, *Biophys. J.* 96 (2009) 2261–2267; Chen et al, *BBA-Bioenergetics* (2017), 1858, 249-258".

Comment 2:

Line 156-157: "The head of Bcr421 in the stromal side interacts with Chl408 at distance of 3.9 Å, and therefore may mediate EET from IsiA to the PSI core."

My simple question here is: How? What does author think by saying “mediate EET”? Do they mean to be physically involved in excitation energy transfer process between IsiA and PSI so we have Chl (IsiA)-Bcr(IsiA)-Chl(PSI) pathway for excitation energy transfer? Or authors are trying to “furnish” this pigment into hypothetical quenching mechanism of PSI by IsiA? If it is correct, this is very, very unlikely to happen (for a multiple reasons).

Author reply 3-2:

Thanks for your comments. According to your comment, we added the Bcr-Chl interactions between PSI and IsiA as supplementary Fig. 8. We added following sentence on lines 190 to 193. “The head of Bcr421 in the stromal side interacts with Chl408 at a distance of 3.9 Å. On the other hand, the opposite head of the Bcr in IsiA2, IsiA3 and IsiA4 interact with Chl817 of PsaA, Chl101 of PsaJ and Chl102 of PsaJ with distances of 6.3, 4.6 and 5.8 Å, respectively. Therefore, they may mediate EET from IsiA to the PSI core.”

Comment 3:

Line 165-221: I think those two paragraphs are exaggerated. All what authors have are pigment-pigment or pigment-protein residue distances based on the molecular structure. Distances are not equal to inter-pigment interactions (the way how pigment molecules interact on each other via energetic coupling, etc.) that could be calculated based on pigment distances and their mutual orientations (electric dipoles). It is rather pigment-ligand framework that shows places where possible, important pigment-pigment interactions may (but not necessary) occur. In this context Figure 4 is pretty much useless as it does not explain anything.

Author reply 3-3:

According to your comments, we calculated dipole orientation factors, Förster energy transfer rate, lifetime and half-life times between adjacent Chls based on the Förster-type energy transfer. These values are summarized in Supplementary Table 4 and 5. Accordingly, the EET pathways were slightly changed and we modified Figs.3e, 4h, 4i and 4j as well as the corresponding text significantly by discussing the EET pathways based on the calculated Förster energy transfer rate.

Comment 4:

Lines: 240 - 274, I have hard time to follow authors understanding that stands behind

performing TRF measurements in this way and their interpretation. Why would authors do fluorescence decay measurements at 77 K for PSI and IsiA-PSI but then compare fluorescence dynamics at 685 nm from room temperature TRF data that are not even shown at all? I thought that PSI has no detectable fluorescence at room temperature so how come it is detectable here. It makes little sense for me.

Just assuming (I guess) that authors tried to somehow reveal reverse excitation energy transfer from PSI to IsiA (in other words PSI quenching by IsiA) it would be most useful to excite PSI Qy band just past 680 nm that minimally overlaps with Qy absorption band of the IsiA and make sure that electrochemistry of the special pair in the PSI is disrupted so potential mechanisms of overexcitation mitigation, like excitation quenching, would trigger. All what is provided in materials and methods is “The light source was a Ti:Sapphire laser (Tsunami, Spectra-Physics, USA).” which is very vague statement without important details. Tsunami oscillator has tuning range between 700 and 1080 nm and it cannot fit in any way to absorption spectrum of IsiA/PSII to be used as excitation source. Provide more details.

Author reply 3-4:

Thanks for your important comments. At 77K, fluorescence is detected from chlorophylls working as energy traps in each complex; therefore, TRF spectra at 77 K give us rich information on inter/intra-complex energy transfer. Actually, Figures 5a and 5b indicate energy transfer from IsiA to PSI and within PSI. However, it still remains unclear from TRF measurements by a time-correlated single photon counting (TCSPC) method at 77 K whether the IsiA-induced quenching occurs at physiological temperature and whether the quenching occurs in very early time region. Therefore, we carried other TRF measurements at room temperature by a fluorescence upconversion (UPC) method, which can be monitored in a femtosecond time range. We believe that a combination of the TCSPC measurements at 77 K and the UPC measurements at room temperature is a powerful method to reveal excitation-energy transfer in photosynthesis. Actually, we have reported several results by the combination of the TRF methods (Nagao et al., *J. Phys. Chem. B* 123, 66&2673; Kim et al., *J. Biol. Chem.* 46, 18951 (2017)).

Moreover, we added detailed information of the UPC performed in the present study to the revised manuscript. The excitation wavelength was 425 nm (the second harmonic of Ti:Sapphire laser), which excites the Soret band of Chl *a* in both IsiA and PSI.

Altogether, we revised the Method section as follows:

Lines 495-501: “Fluorescence rise and decay curves were measured with a femtosecond fluorescence upconversion method^{ref}. The excitation pulse was the second harmonic (425 nm) of a Ti:sapphire laser (Tsunami, Spectra-Physics, USA) (80 MHz) pumped with a diode-pumped solid state laser (Millennia Xs, Spectra-Physics, USA). To avoid polarization effects, the angle between the polarizations of the excitation and probe beams was set to the magic angle by a $\lambda/2$ plate. The instrumental response function had a pulse width of ~ 200 fs.”.

Ref: Akimoto & Mimuro, Photochem Photobiol 83, 163-170, (2007)

Minor points:

- Improve language. There are apparent mistakes in many places: e.g. – line 44 “...harvesting solar energy and transferring them...”, line 64 “donoation”, line 178 “extenisvly”, etc.

Author reply 3-5:

We modified them; thank you very much.

- Line 72, 101, and so on: IsiA subunits – IsiA is not a subunit of something nor has any subunits itself (like PsaA, B,... in PSI). It is just a protein complex.

Author reply 3-6:

We removed “subunit” or changed to “IsiA protein”, according to your comments.

- Line 137, and on vs Figure 2 caption: In the main text and in Figure 2 authors use ChlXXX notation while in the Figure 2 caption CLXXX is used. Decide which one you would like to use.

Author reply 3-7:

We changed to ChlXXX throughout the text.

- Line 250: “...the IsiA molecule” – IsiA is not a molecule. Molecule is group of atoms chemically bound. As IsiA consists of protein and pigments that are not chemically bond so IsiA is not a molecule.

Author reply 3-8:

We removed “molecule” according to your suggestion.

- Figure 2b: it is very congested and will be extremely difficult to see anything on it in the printed version

Author reply 3-9:

Thanks for your comments. To reduce the congestion, we divided the figure into Chl and Bcr. The figure for the arrangement of the Bcrs was added to the middle panel. Furthermore, we enlarged the figures and decreased the arrows.

- Figure 3 d and c: not the best choice of colors to mark Chls and Bcrs. It is extremely difficult to see what is what as Bcrs and Chls are in the same color.

Author reply 3-10:

According to your comments, we changed the colors in figure 3c and 3d.

- Figure 5: add legends to a-c to improve figure clarity (black –PSI and red-IsiA-PSI)

Author reply 3-11:

According to your comments, we described the legends in more detail.

Reviewers' comments:

Reviewer #1 (Remarks to the Author):

All points raised have been addressed in a satisfying way.
Page 7, line 126: The words "the" are consecutive.

Reviewer#2 (Remarks to the Author):

I've looked over the revised manuscript and I saw that my remarks regarding model quality were not addressed at all. The statistics cited in the table are identical to ones in the previous submission. If the authors do not wish to change this they should supply the full pdb validation report.

Reviewer #3 (Remarks to the Author):

Thank you for detailed responses to my comments and questions. I carefully read them as well the revised manuscript too. I am completely satisfied with authors' responses, corrections and improvements done in the manuscript.

Dariusz Niedzwiedzki, Ph.D.

Responses to reviewers' comments

Responses to Reviewer #1

Comment 1-1:

1. All points raised have been addressed in a satisfying way.

Page 7, line 126: The words "the" are consecutive.

Author reply 1-1:

We modified it. Thank you.

Responses to Reviewer #2

Comment 2-1:

I've looked over the revised manuscript and I saw that my remarks regarding model quality were not addressed at all. The statistics cited in the table are identical to ones in the previous submission. If the authors do not wish to change this they should supply the full pdb validation report.

Author reply 2-1:

We apologize for missing to answer your previous comment regarding the model quality. The statistics in the table were not changed, so we provide the full pdb validation report here. We also added the following sentences in the revised manuscript to explain the model quality (Page 6 Line 101-106).

"The local resolution map was shown in Supplementary Fig. 3c. The peripheral regions of PSI-IsiA have relatively low resolution, resulting in rather high numbers of clashscore and poor rotamers in the final refined structure (Supplementary Table 1). However, the Chl and Bcr molecules, which were related to energy pathways, have well defined densities and therefore were completely assigned."

Responses to Reviewer #3

Comment 3-1:

Thank you for detailed responses to my comments and questions. I carefully read them as well the revised manuscript too. I am completely satisfied with authors' responses, corrections and improvements done in the manuscript.

Author reply 3-1:

Thank you for your careful reading and evaluation.